# A Decade's Battle on Dataset Bias: Are We There Yet?

**Zhuang Liu**    **Kaiming He**[*]
Meta AI Research, FAIR

## Abstract

We revisit the "dataset classification" experiment suggested by Torralba & Efros (2011) a decade ago, in the new era with large-scale, diverse, and hopefully less biased datasets as well as more capable neural network architectures. Surprisingly, we observe that modern neural networks can achieve excellent accuracy in classifying which dataset an image is from: *e.g.*, we report 84.7% accuracy on held-out validation data for the three-way classification problem consisting of the YFCC, CC, and DataComp datasets. Our further experiments show that such a dataset classifier could learn semantic features that are generalizable and transferable, which cannot be explained by memorization. We hope our discovery will inspire the community to rethink issues involving dataset bias.

## 1 Introduction

In 2011, Torralba & Efros (2011) called for a battle against dataset bias in the community, right before the dawn of the deep learning revolution (Krizhevsky et al., 2012). They introduced the "*Name That Dataset*" experiment, where images are sampled from a dataset and a model is trained on the union of these images to classify which dataset an image is taken. Remarkably, datasets at that time could be classified with high accuracy. They also found that a model trained on one dataset can only perform well on that dataset but fails to generalize to others.

In response to this, over the decade that followed, progress on building diverse, large-scale, comprehensive, and hopefully less biased datasets (Lin et al., 2014; Russakovsky et al., 2015; Thomee et al., 2016; Kuznetsova et al., 2020; Schuhmann et al., 2022) has been an engine powering the deep learning revolution, especially in the pre-training era. In parallel, advances in algorithms, particularly neural network architectures, have achieved unprecedented levels of ability on discovering concepts, abstractions, and patterns—including *bias*—from data.

In this work, we take a renewed "*unbiased look at dataset bias*" (Torralba & Efros, 2011) after the decade-long battle. Our study is driven by the tension between building less biased datasets versus developing more capable models—the latter was less prominent at the time of Torralba & Efros (2011). While efforts to reduce bias in data may lead to progress, the development of advanced models could better exploit dataset bias and thus counteract the promise.

Our study is based on a fabricated task we call *dataset classification*, which is the "*Name That Dataset*" experiment designed in Torralba & Efros (2011) (Figure 1). The datasets we experiment with are presumably among the most diverse, largest, and uncurated datasets in the wild, collected from the Internet. For example, a typical combination we study, referred to as "YCD", consists of images from YFCC (Thomee et al., 2016), CC (Changpinyo et al., 2021), and DataComp (Gadre et al., 2023) and presents a 3-way dataset classification problem.

To our (and many of our initial readers') surprise, modern neural networks can achieve excellent accuracy on such a dataset classification task. Trained in the aforementioned YCD set that is challenging for human beings (Figure 1), a model can achieve >84% classification accuracy on the *held-out* validation data, *vs*. 33.3% of chance-level guess. This observation is highly robust, over a large variety of dataset combinations and across different generations of architectures (Krizhevsky et al., 2012; Simonyan & Zisserman, 2015; He et al., 2016; Dosovitskiy et al., 2021; Liu et al., 2022), with very high accuracy (*e.g.*, over 80%) achieved in most cases.

---

[*]Work done at Meta; now at MIT. Code: github.com/liuzhuang13/bias

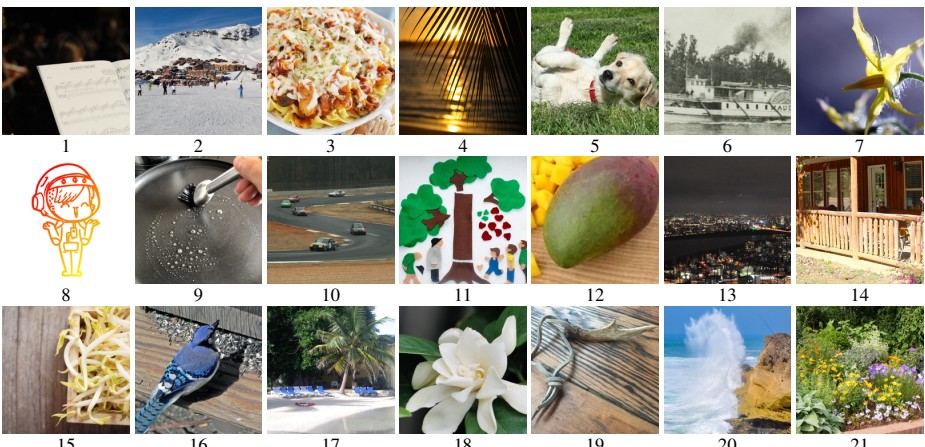

Figure 1: **The "Name That Dataset" game (Torralba & Efros, 2011) in 2024**: These images are sampled from three modern datasets: YFCC (Thomee et al., 2016), CC (Changpinyo et al., 2021), and DataComp (Gadre et al., 2023). *Can you specify which dataset each image is from*? While these datasets appear to be less biased, we discover that neural networks can easily accomplish this "*dataset classification*" task with surprisingly high accuracy on the held-out validation set.

Answer: YFCC: 1, 4, 7, 10, 13, 16, 19; CC: 2, 5, 8, 11, 14, 17, 20; DataComp: 3, 6, 9, 12, 15, 18, 21

For such a dataset classification task, we have a series of observations that are analogous to those observed in *semantic* classification tasks (*e.g.*, object classification). For example, we observe that training the dataset classifier on *more* samples, or using *stronger* data augmentation, can *improve* accuracy on held-out validation data, even though the training task becomes harder. This is similar to the generalization behavior in semantic classification tasks. This behavior suggests that the neural network attempts to discover dataset-specific patterns—a form of bias–to solve the dataset classification task. Further experiments suggest that the representations learned by classifying datasets carry some semantic information that is transferrable to image classification tasks.

As a comparison, if the samples of different datasets were unbiasedly drawn from the same distribution, the model should not discover any dataset-specific bias. To check this, we study a pseudo-dataset classification task, in which the different "datasets" are uniformly sampled from a single dataset. We observe that this classification task quickly becomes intractable, as the only way for the classifier to approach this task is to memorize every single instance and its subset identity. As a result, increasing the number of samples, or using stronger data augmentation, makes memorization more difficult or intractable in experiments. No transferability is observed. These behaviors are strikingly contrary to those of the real dataset classification task.

More surprisingly to us, we observe that *self-supervised* learning models are also highly capable of capturing certain bias among different datasets. Specifically, we pre-train a self-supervised model on the union of different datasets, without using any dataset identity as the labels. Then with the pre-trained representations frozen, we train a linear classifier for the dataset classification task. Although this linear layer is the only layer that is tunable by the dataset identity labels, the model can still achieve a high accuracy (*e.g.*, 78%) for dataset classification. This *transfer learning* behavior resembles the behaviors of typical self-supervised learning methods (*e.g.*, for image classification).

In summary, we report that modern neural networks are surprisingly capable of discovering hidden bias from different datasets. This observation is true even for modern datasets that are very large, diverse, less curated, and presumably less biased. The neural networks can solve this task by discovering generalizable patterns (*i.e.*, generalizable from training data to validation data, or to downstream tasks), exhibiting behaviors analogous to those observed in semantic classification tasks. Comparing with the game of "*Name That Dataset*" in Torralba & Efros (2011) a decade ago, this game even becomes way easier given today's capable neural networks. In this sense, the issue involving dataset bias has not been relieved. There is still a question about how representative our current pre-training datasets are of the real world, and furthermore, how much more generalizable models could become by building more diverse and less biased training datasets. We hope our discovery will stimulate discussions in the community regarding dataset bias in this new era.

## 2 A BRIEF HISTORY OF DATASETS

**Pre-dataset Eras.** The concept of "datasets" did not emerge directly out of the box in the history of computer vision research. Before the advent of computers (*e.g.*, see Helmholtz's book of the 1860s (Von Helmholtz, 1867)), scientists had already recognized the necessity of "test samples", often called "stimuli" back then, to examine their computational models about the human vision system. The stimuli often consisted of synthesized patterns, such as lines, stripes, and blobs. The practice of using synthesized patterns was followed in early works on computer vision.

Immediately after the introduction of devices for digitizing photos, researchers were able to *validate* and justify their algorithms on one or very few real-world images (Roberts, 1963). For example, the *Cameraman* image (Schreiber, 1978) has been serving as a standard test image for image processing research since 1978. The concept of using data (which was not popularly referred to as "datasets") to *evaluate* computer vision algorithms was gradually formed by the community.

**Datasets for Task Definition.** With the introduction of *machine learning* methods into the computer vision community, the concept of "datasets" became clearer. In addition to the data for the validation purpose, the application of machine learning introduced the concept of *training data*, from which the algorithms can optimize their model parameters.

As such, the training data and validation data put together inherently *define a task* that is of interest. For example, the MNIST dataset (LeCun et al., 1998) defines a 10-digit classification task; the Caltech-101 dataset (Fei-Fei et al., 2004) defines an image classification task of 101 object categories; the PASCAL VOC suite of datasets (Everingham et al., 2010) define a family of classification, detection, and segmentation tasks of 20 object categories.

To incentivize more capable algorithms, more challenging tasks were defined. The most notable example of this kind, in today's context, is the ImageNet dataset (Deng et al., 2009). ImageNet has over one million images defined with 1000 classes (many of them being fine-grained animal species), which is nontrivial even for normal human beings to recognize (Karpathy, 2014). At the time when ImageNet was proposed, algorithms for solving this *task* appeared to be cumbersome— *e.g.*, the organizers provided SIFT features (Lowe, 2004) pre-computed to facilitate studying this problem, and typical methods back then may train 1000 SVM classifiers, which in itself is a nontrivial problem (Vedaldi & Zisserman, 2012). Hypothetically, if ImageNet was to remain as a task on its own, like many previous popular datasets, we wouldn't be able to witness the deep learning revolution.

But a paradigm shift awaited.

**Datasets for Representation Learning.** Right after the deep learning revolution in 2012 (Krizhevsky et al., 2012), the community soon discovered that the neural network representations learned on large-scale datasets like ImageNet are *transferrable* (Donahue et al., 2014; Girshick et al., 2014; Yosinski et al., 2014). The discovery brought in a paradigm shift in computer vision: it became a common practice to pre-train representations on ImageNet and transfer them to downstream tasks.

As such, the ImageNet dataset was no longer a task of its own; it became a pinhole of the *universal visual world* that we want to represent. Thus, the used-to-be cumbersome aspects became advantages of this dataset: a larger number of images and more diversified categories than most other datasets at that time, and empirically these properties turned out to be important for learning good representations.

Encouraged by ImageNet's enormous success, the community began to pursue more general and ideally universal visual representations. Tremendous effort has been paid on building larger, more diversified, and hopefully less biased datasets. Examples include YFCC100M (Thomee et al., 2016), CC12M (Changpinyo et al., 2021), and DataComp-1B (Gadre et al., 2023)—the main datasets we study in this paper—among many others (Sun et al., 2017; Desai et al., 2021; Srinivasan et al., 2021; Schuhmann et al., 2022). It is intriguing to notice that the building of these datasets does *not* always define a task of interest to solve; actually, many of these large-scale datasets do not provide a split of training / validation sets. It is with the goal of *pre-training* in mind that these datasets were built.

## 3 ON DATASET BIAS

Given the increasing importance of datasets, the bias introduced by datasets has drawn the community's attention. Torralba & Efros (2011) presented the dataset classification problem and examined

dataset bias in the context of hand-crafted features with SVM classifiers. Tommasi et al. (2015) studied the dataset classification problem using neural networks, specifically focusing on linear classifiers with pre-trained ConvNet features (Donahue et al., 2014). The datasets they studied are smaller in scale and simpler comparing with today's web-scale data.

The concept of classifying different datasets has been further developed in domain adaption methods (Tzeng et al., 2014; Ganin et al., 2016). These methods learn classifiers to adversarially distinguish features from different domains, where each domain can be thought of as a dataset. The problems studied by these methods are known to have significant domain gaps. On the contrary, the datasets we study are presumably less distinguishable, at least for human beings.

Another direction on studying dataset bias is to replicate the collection process of a dataset and examine the replicated data. ImageNetV2 (Recht et al., 2019) replicated the ImageNet validation set's protocol. It observed that this replicated data still clearly exhibits bias as reflected by accuracy degradation. The bias is further analyzed in (Engstrom et al., 2020).

Many benchmarks (Hendrycks & Dietterich, 2018; Zendel et al., 2018; Koh et al., 2021; Hendrycks et al., 2021) have been created for testing models' generalization under various forms of biases, such as common corruptions and hazardous conditions. There is also a rich line of work on mitigating dataset bias. Training on multiple datasets (Lambert et al., 2020; Nguyen et al., 2022) can potentially mitigate dataset bias. Methods that adapt models to data with different biases at test time (Sun et al., 2020; Wang et al., 2021) have also gained popularity recently.

**Different Notions of Bias.** It is worth noting that this study's focus is the bias among multiple datasets (hence "dataset" bias, instead of "data" bias). This mostly concerns the proper coverage of concepts and objects, or in other words, how representative the dataset is for the real world. It is not to be confused with another common notion of bias in data—social and stereotypical bias. This notion concerns more on algorithmic fairness (Mitchell et al., 2021) and could be found within a single dataset, *e.g.*, gender or race bias. These two notions are related but emphasize different aspects. For example, a simple dataset of indoor furnitures is mostly free of social bias, but is extremely biased in terms of representativeness of the world.

Addressing social bias in data is an active area of research. Several well-known datasets have been identified with biases in demographics (Buolamwini & Gebru, 2018; Yang et al., 2020) and geography (Shankar et al., 2017). They also contain harmful societal stereotypes (van Miltenburg, 2016; Prabhu & Birhane, 2021; Birhane et al., 2021; Zhao et al., 2021). Addressing these biases is critical for fairness and ethical considerations. Tools like REVISE (Wang et al., 2022) and Know Your Data (Google People + AI Research, 2021) offer automatic analysis for potential bias in datasets. Debiasing approaches, such as adversarial learning (Zhang et al., 2018a) and domain-independent training (Wang et al., 2020), have also shown promise in reducing the effects of dataset bias.

## 4 DATASET CLASSIFICATION

The dataset classification task (Torralba & Efros, 2011) is defined like an image classification task, but each dataset forms its own class. It creates an $N$-way classification problem where $N$ is the number of datasets. The accuracy is evaluated on a held-out validation set sampled from these datasets.

### 4.1 ON THE DATASETS WE USE

We intentionally choose the datasets that can make the dataset classification task challenging. We choose our datasets based on the following considerations: (1) They are large in scale. Smaller datasets might have a narrower range of concepts covered, and they may not have enough training images for dataset classification. (2) They are general and diversified. We avoid datasets that are about a specific scenario (*e.g.*, cities (Cordts et al., 2016), scenes (Zhou et al., 2017)) or a specific meta-category of objects (*e.g.*, flowers (Nilsback & Zisserman, 2008), pets (Parkhi et al., 2012)). (3) They are collected as with the intention of pre-training generalizable representations, or have been used with this intention. We emphasize the difference between the "pre-training" datasets and "benchmark" datasets here, as it is more accepted that the evaluation benchmark datasets are often unique and biased (Raji et al., 2021; Koch et al., 2021). Based on these criteria, we choose the datasets listed in Table 1.

| dataset | description |
|---|---|
| YFCC (Thomee et al., 2016) | 100M Flickr images |
| CC (Changpinyo et al., 2021) | 12M Internet image-text pairs |
| DataComp (Gadre et al., 2023) | 1B image-text pairs from Common Crawl |
| WIT (Srinivasan et al., 2021) | 11.5M Wikipedia images-text pairs |
| LAION (Schuhmann et al., 2022) | 2B image-text pairs from Common Crawl |
| ImageNet (Deng et al., 2009) | 14M images from search engines |

Table 1: Datasets used in our experiments.

Although these datasets are supposedly more diverse, there are still differences in their collection processes that potentially contribute to their individual biases. For example, their sources are different: Flickr is a website where users upload and share photos, Wikipedia is a website focused on knowledge and information, Common Crawl is an organization that crawls the web data, and the broader Internet involves a more general range of content than these specific websites. Moreover, different levels of curation have been involved in the data collection process: *e.g.*, LAION was collected by reverse-engineering the CLIP model (Radford et al., 2021) and reproducing its zero-shot accuracy (Schuhmann et al., 2022).

Despite our awareness of these potential biases, a neural network's excellent ability to capture them is beyond our expectation. In particular, we note that we evaluate a network's dataset classification accuracy by applying it to each validation image *individually*, which ensures that the network has no opportunity to exploit the underlying statistics of several images.

## 4.2 MAIN OBSERVATION

We observe surprisingly high accuracy achieved by neural networks in this dataset classification task. This observation is robust across different settings. By default, we randomly sample 1M and 10K images from each dataset as training and validation sets, respectively. We train a ConvNeXt-T model (Liu et al., 2022) following common practice of supervised training (implementation details are in Appendix B).

| YFCC | CC | DataComp | WIT | LAION | ImageNet | accuracy |
|:---:|:---:|:---:|:---:|:---:|:---:|:---:|
| ✓ | ✓ | ✓ | | | | 84.7 |
| ✓ | ✓ | | ✓ | | | 83.9 |
| ✓ | ✓ | | | ✓ | | 85.0 |
| ✓ | ✓ | | | | ✓ | 92.7 |
| ✓ | | ✓ | ✓ | | | 85.8 |
| ✓ | | ✓ | | ✓ | | 72.1 |
| ✓ | | ✓ | | | ✓ | 90.2 |
| ✓ | | | ✓ | ✓ | | 86.6 |
| ✓ | | | ✓ | | ✓ | 86.7 |
| ✓ | | | | ✓ | ✓ | 91.9 |
| | ✓ | ✓ | ✓ | | | 83.6 |
| | ✓ | ✓ | | ✓ | | 62.8 |
| | ✓ | ✓ | | | ✓ | 82.8 |
| | ✓ | | ✓ | ✓ | | 84.3 |
| | ✓ | | ✓ | | ✓ | 91.3 |
| | ✓ | | | ✓ | ✓ | 84.1 |
| | | ✓ | ✓ | ✓ | | 71.5 |
| | | ✓ | ✓ | | ✓ | 88.9 |
| | | ✓ | | ✓ | ✓ | 68.2 |
| | | | ✓ | ✓ | ✓ | 90.7 |
| ✓ | ✓ | ✓ | | | | 84.7 |
| ✓ | ✓ | ✓ | ✓ | | | 79.1 |
| ✓ | ✓ | ✓ | ✓ | ✓ | | 67.4 |
| ✓ | ✓ | ✓ | ✓ | ✓ | ✓ | 69.2 |

Table 2: **Dataset classification yields high accuracy in all combinations. Top panel**: all 20 combinations that involve 3 datasets out of all 6. **Bottom panel**: combinations with 3, 4, 5, or 6 datasets. All results are with 1M training images sampled from each dataset.

We observe the following behaviors in our experiments:

**High accuracy is observed across *dataset combinations*.** In Table 2 (top panel), we enumerate all 20 ($C_6^3$) possible combinations of choosing 3 out of the 6 datasets listed in Table 1. In summary, in all cases, the network achieves >62% dataset classification accuracy; and in 16 out of all 20 combinations, it even achieves >80% accuracy. In the combination of YFCC, CC, and ImageNet, it achieves the highest accuracy of **92.7%**. Note that the chance-level guess gives 33.3% accuracy.

In Table 2 (bottom panel), we study combinations involving 3, 4, 5, and all 6 datasets. As expected, using more datasets leads to a more difficult task, reflected by the decreasing accuracy. However, the network still achieves 69.2% accuracy when all 6 datasets are included. The confusion matrix of the 6-way classifier can be found in Appendix C.

**High accuracy is observed across *model architectures*.** Table 3 shows the results on YCD using different generations of models: AlexNet (Krizhevsky et al., 2012), VGG (Simonyan & Zisserman, 2015), ResNet (He et al., 2016), ViT (Dosovitskiy et al., 2021), and ConvNeXt (Liu et al., 2022).

We observe that *all architectures can solve the task excellently*: 4 out of the 5 networks achieve excellent accuracy of >80%, and even the classical AlexNet achieves a strong result of 77.8%.

This result shows the neural networks are extremely good at capturing dataset biases, regardless of their concrete architectures. There has been significant progress in network architecture design after the AlexNet paper, including normalization layers (Ioffe & Szegedy, 2015;

| model | accuracy |
|---|---|
| AlexNet | 77.8 |
| VGG-16 | 83.5 |
| ResNet-50 | 83.8 |
| ViT-S | 82.4 |
| ConvNeXt-T | 84.7 |

Table 3: **Different model architectures all achieve high accuracy**. Results are on the YCD combination with 1M images each.

Ba et al., 2016), residual connections (He et al., 2016), self-attention (Vaswani et al., 2017; Dosovitskiy et al., 2021). The "inductive bias" in network architectures can also be different (Dosovitskiy et al., 2021). Nevertheless, none of them appears to be indispensable for dataset classification (*e.g.*, VGG (Simonyan & Zisserman, 2015) has none of these components): the ability to capture dataset bias may be inherent in deep neural networks, rather than enabled by specific components.

**High accuracy is observed across different *model sizes*.** By default, we use ConvNeXt-Tiny (27M parameters) (Liu et al., 2022). The term "Tiny" is with reference to the modern definition of ViT sizes (Touvron et al., 2021; Dosovitskiy et al., 2021) and is comparable to ResNet-50 (25M) (He et al., 2016). In Figure 2, we report results of models with different sizes by varying widths and depth.

To our further surprise, even *very small* models can achieve strong accuracy for the dataset classification task. A ConvNeXt with as few as 7K parameters (3/10000 of ResNet-50) achieves 72.4% accuracy on classifying YCD. This suggests that neural networks' structures are very effective in learning the underlying dataset biases. Dataset classification can be done without a massive number of parameters, which is often credited for deep learning's success in conventional visual recognition.

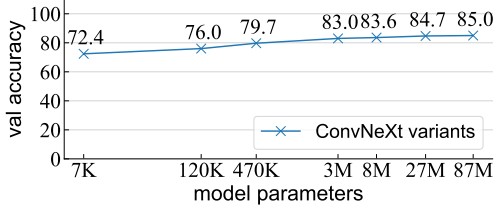

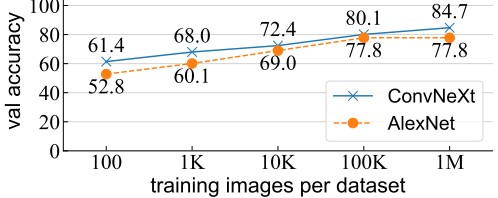

Figure 2: **Models of different sizes all achieve very high accuracy**, while they can still be substantially smaller than the sizes of typical modern networks. Here the models are variants of ConvNeXt (Liu et al., 2022), whose "Tiny" size has 27M parameters. Results are on YCD combination with 1M training images from each set.

Figure 3: **Dataset classification accuracy increases with the number of training images**. This behavior suggests that the model is learning certain patterns that are generalizable, which resembles the behavior observed in typical semantic classification tasks. Results are on YCD, with each model trained for the same iterations.

We also observe that larger models get increasingly better, although the return becomes diminishing. This is consistent with observations on conventional visual recognition tasks. Moreover, we have not observed overfitting behaviors to the extent of the model sizes and dataset scales we have studied. This implies that there may exist generalizable patterns that help the models determine dataset identities and the model is not trying to memorize the training data. More investigations on generalization and memorization are presented next.

**Dataset classification accuracy benefits from *more training data*.** We vary the number of training images for YCD classification and present results in Figure 3.

Intriguingly, models trained with *more* data achieve *higher* validation accuracy. This trend is consistently observed in both the modern ConvNeXt and the classical AlexNet. While this behavior appears to be natural in *semantic* classification tasks, we remark that this is not necessarily true in dataset classification: in fact, if the models *were* focusing on *memorizing* the training data, their generalization performance on the validation data might decrease. The observed behavior—*i.e.*, more training data improves validation accuracy—suggests that the model is learning certain semantic patterns that are generalizable to unseen data, rather than memorizing or overfitting the training data.

**Dataset classification accuracy benefits from *data augmentation*.** Data augmentation (Krizhevsky et al., 2012) is expected to have similar effects as increasing the dataset size (which is the rationale behind its naming). Our default training setting uses random cropping (Szegedy et al., 2015), RandAug (Cubuk et al., 2020), MixUp (Zhang et al., 2018b), and CutMix (Yun et al., 2019) as data augmentations. Table 4 shows the results of using reduced or no data augmentations.

| augmentation / training images per dataset | 10K | 100K | 1M |
|---|---|---|---|
| no aug | 43.2 | 71.9 | 76.8 |
| w/ RandCrop | 66.1 | 74.5 | 84.2 |
| w/ RandCrop, RandAug | 70.2 | 78.0 | 85.0 |
| w/ RandCrop, RandAug, MixUp / CutMix | 72.4 | 80.1 | 84.7 |

Table 4: **Data augmentation improves dataset classification accuracy**, similar to the behavior of semantic classification tasks. Results are on the YCD combination.

Adding data augmentation makes it more difficult to memorize the training images, while we observe that using stronger data augmentation consistently *improves* the dataset classification accuracy. This behavior remains largely consistent regardless of the number of training images per dataset. Again, this behavior mirrors that observed in *semantic* classification tasks, suggesting that dataset classification is approached not through memorization, but by learning patterns that are generalizable from the training set to the unseen validation set.

**Summary.** In sum, we have observed that neural networks are highly capable of solving the dataset classification task with good accuracy. This observation holds true across a variety of conditions, including different combinations of datasets, various model architectures, different model sizes, dataset sizes, and data augmentation strategies. In comparison, we report human performance on a similar task in Appendix A.

## 5 ANALYSIS

In this section, we analyze the model behaviors in different modified versions involving the dataset classification task. This reveals more intriguing properties of neural networks for dataset classification.

### 5.1 LOW-LEVEL SIGNATURES?

There is a possibility that the high accuracy is simply due to low-level signatures, which are less noticeable to humans but are easily identifiable by neural networks. Potential signatures could involve JPEG compression artifacts (*e.g.*, different datasets may have different compression quality factors) and color quantization artifacts (*e.g.*, colors are trimmed or quantized depending on the individual dataset). We design a set of experiments that help us preclude this possibility.

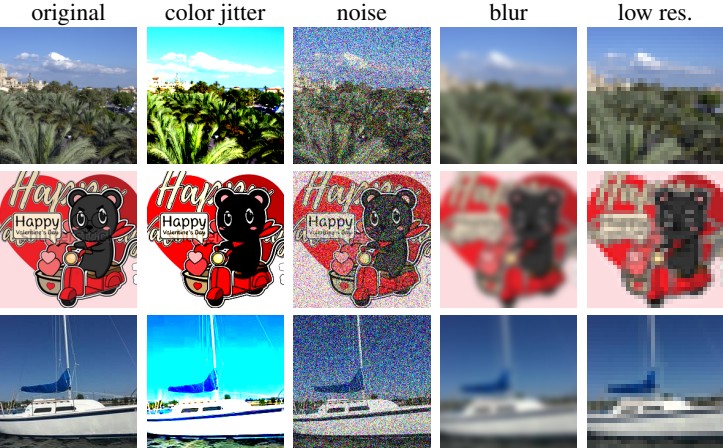

Figure 4: **Different corruptions for suppressing low-level signatures**. We apply a certain type of corruption to both the training and validation sets, on which we train and evaluate our model.

Specifically, we apply a certain type of image corruption to both the training and validation sets, on which we train and evaluate our model. In other words, we perform the dataset classification task *on corrupted data*.[1] We consider four types of image corruption: (i) color jittering (Krizhevsky et al., 2012), (ii) adding Gaussian noise with a fixed standard deviation; (iii) blurring the image by a fixed-size Gaussian kernel; and (iv) reducing the image resolution. Figure 4 shows examples for each corruption. Note that we apply one type of corruption each time.

Table 5 shows the dataset classification results for each image corruption. As expected, corruption reduces the classification accuracy, as both training and validation sets are affected. Despite degradation, strong classification accuracy can still be achieved, especially when the degree of corruption is weaker. Introducing these different types of corruption should effectively disrupt low-level signatures, such as JPEG or color quantization artifacts. The results imply that the models attempt to solve the dataset classification task beyond using low-level biases.

| corruption (on train+val) | accuracy |
|---|---|
| none | 84.7 |
| color jittering (strength: 1.0) | 81.1 |
| color jittering (strength: 2.0) | 80.2 |
| Gaussian noise (std: 0.2) | 77.3 |
| Gaussian noise (std: 0.3) | 75.1 |
| Gaussian blur (radius: 3) | 80.9 |
| Gaussian blur (radius: 5) | 78.1 |
| low resolution (64×64) | 78.4 |
| low resolution (32×32) | 68.4 |

Table 5: **High accuracy are achieved on different corrupted versions of the dataset classification task**. This suggests that low-level signature is not a main responsible factor. Results are on the YCD combination.

| imgs per set | w/o aug | w/ aug |
|---|---|---|
| 100 | 100.0 | 100.0 |
| 1K | 100.0 | 100.0 |
| 10K | 100.0 | fail |
| 100K | fail | fail |

Table 6: *Training* **accuracy on a pseudo-dataset classification task.** Here we create 3 pseudo-datasets, all of which are sampled without replacement from the same source dataset (YFCC). This *training* task is more difficult for the network to solve if given more training images and/or stronger data augmentation. Validation accuracy is ∼33% as no transferrable pattern is learned.

## 5.2 MEMORIZATION OR GENERALIZATION?

In Sec. 4.2, we have shown that the models learned for dataset classification behave like those learned for semantic classification tasks (Figure 3 and Table 4), since they exhibit *generalization* behaviors. This behavior is in sharp contrast with *memorization* behavior, as we discuss in the next comparison.

---

[1]This is different from *data augmentation*, which applies random image corruption to the training data.

We consider a *pseudo*-dataset classification task. In this scenario, we manually create multiple pseudo-datasets, all of which are sampled without replacement from the same source dataset. We expect this process to give us multiple pseudo-datasets that are truly unbiased.

Table 6 reports the *training* accuracy of a model trained for this pseudo-dataset classification task, using different numbers of training images per set, without *vs.* with data augmentation. When the task is relatively simple, the model achieves 100% training accuracy; however, when the task becomes more difficult (more training images or stronger augmentation), the model fails to converge, as reflected by unstable, non-decreasing loss curves.

This phenomenon implies that the model attempts to *memorize* individual images and their labels to accomplish this pseudo-dataset classification task. Because the images in these pseudo-datasets are unbiased, there should be no shared patterns that can be discovered to discriminate these different sets. As a result, the model is forced to memorize the images and their random labels, similar to the scenario in Zhang et al. (2017). But memorization becomes more difficult when given more training images or stronger augmentation, which fails the training process after a certain point.

This phenomenon is unlike what we have observed in our real dataset classification task (Figure 3 and Table 4). This again suggests that the model attempts to capture shared, generalizable patterns in the real dataset classification task.

Although it may seem evident, we note that the model trained for the pseudo-dataset classification task does *not* generalize to validation data (which is held out and sampled from each pseudo-dataset). Even when the training accuracy is 100%, we report a chance-level accuracy of ~33% in the validation set.

## 5.3 Self-supervised Learning

Thus far, all our dataset classification results are presented under a *fully-supervised* protocol: the models are trained end-to-end with full supervision. Next, we explore a *self-supervised* protocol, following the common protocol used for semantic classification tasks in self-supervised learning.

Formally, we pre-train a self-supervised learning model MAE (He et al., 2022) without using any labels. Then we freeze the features extracted from this pre-trained model, and train a linear classifier using supervision for the dataset classification task. This is referred to as the linear probing protocol. We note that in this protocol, *only the linear classifier layer is tunable* under the supervision of the dataset classification labels. Linear probing presents a more challenging scenario.

Table 7 shows the results under the self-supervised protocol. Even with MAE pre-trained on standard ImageNet (which involves *no* YCD images), the model achieves 76.2% linear probing accuracy for dataset classification. In this case, only the linear classifier layer is exposed to the classification data.

Using MAE pre-trained on the same YCD training data, the model achieves higher accuracy of 78.4% in linear probing. Note that although this MAE is pre-trained on the same target data, it has no prior knowledge that the goal is for dataset classification. Nevertheless, the pre-trained model can learn features that are more discriminative (for this task) than those pre-trained on the different dataset of ImageNet. This transfer learning behavior again resembles those seen in semantic classification tasks.

| case | accuracy |
|---|---|
| fully-supervised | 82.9 |
| *linear probing w/* | |
| MAE trained on IN-1K | 76.2 |
| MAE trained on YCD | 78.4 |

Table 7: **Self-supervised pre-training, followed by linear probing, achieves high accuracy for dataset classification.** Here, we study MAE (He et al., 2022) as our self-supervised pre-training baseline, which uses ViT-B as the backbone. The fully-supervised baseline for dataset classification is with the same ViT-B architecture (82.9%). Results are on the YCD combination.

| case | transfer acc |
|---|---|
| random weights | 6.7 |
| Y+C+D | 27.7 |
| Y+C+D+W | 34.2 |
| Y+C+D+W+L | 34.2 |
| Y+C+D+W+L+I | 34.8 |
| MAE (He et al., 2022) | 68.0 |
| MoCo v3 (Chen et al., 2021) | 76.7 |

Table 8: **Features learned by classifying datasets can achieve nontrivial results under the linear probing protocol.** Transfer learning (linear probing) accuracy is reported on ImageNet-1K, using ViT-B as the backbone in all entries. The acronyms follow Table 2.

## 5.4 FEATURES LEARNED BY CLASSIFYING DATASETS

We have shown that models trained for dataset classification can well generalize to unseen validation data. Next we study how well these models can be transferred to semantic classification tasks. To this end, we now consider dataset classification as a pretext task, and perform linear probing on the frozen features on a semantic classification task (ImageNet-1K classification). Table 8 shows the results of our dataset classification models pre-trained using different combinations of datasets.[2]

Comparing with the baseline of using random weights, the dataset classification models can achieve non-trivial ImageNet-1K linear probing accuracy. Importantly, using a combination of more datasets can increase the linear probing accuracy, suggesting that *better features are learned by discovering the dataset biases across more datasets.*

As a reference, it should be noted the features learned by dataset classification are significantly worse than those learned by self-supervised learning methods, such as MAE (He et al., 2022) and MoCo v3 (Chen et al., 2021), which is as expected. Nevertheless, our experiments reveal that *the dataset bias discovered by neural networks is relevant to semantic features that are useful for image classification.*

## 5.5 CROSS-DATASET GENERALIZATION

Torralba & Efros (2011) observed that models often struggle to generalize across different datasets. For instance, a model trained on dataset A to recognize cars may perform well on held-out images from dataset A but poorly on a dataset B. We revisit this cross-generalization experiment using the modern and large-scale datasets we study. Due to the lack of a common task defined on them, we use contrastive learning (MoCo v3) (Chen et al., 2021) as a surrogate task and report their validation losses. More result with masked autoencoding (MAE) (He et al., 2022) is available in Appendix C.

| train / eval | YFCC | CC | DataComp | WIT | LAION | ImageNet | average |
|---|---|---|---|---|---|---|---|
| YFCC | **1.761** | 2.202 | 2.668 | 2.083 | 2.764 | 2.026 | 2.251 |
| CC | 1.971 | **1.759** | 1.885 | 2.012 | 1.874 | 1.970 | 1.912 |
| DataComp | 2.216 | 1.891 | **1.772** | 2.161 | 1.801 | 2.023 | 1.977 |
| WIT | 1.969 | 2.059 | 2.238 | **1.742** | 2.288 | 2.004 | 2.050 |
| LAION | 2.332 | 1.902 | 1.787 | 2.236 | **1.779** | 2.097 | 2.022 |
| ImageNet | 1.941 | 2.077 | 2.040 | 2.157 | 2.233 | **1.742** | 2.032 |
| combined | 1.940 | 1.841 | 1.822 | 1.915 | 1.847 | 1.860 | **1.871** |

Table 9: **Cross-dataset generalization with MoCo v3 validation losses. Bold** indicates the lowest for each evaluation dataset (column). A clear diagonal indicates that cross-dataset transfer always has a gap with training on the same dataset. Combining all datasets yields the best result when averaged.

Table 9 shows the results. We can see a clear diagonal with the lowest validation loss in each column. This indicates that on any particular validation dataset, only pre-training on the same training dataset can achieve the best generalization performance. Therefore, despite larger and more diversified datasets, cross-dataset generalization remains a problem. Interestingly, simply combining all datasets (controlling the total number of images by taking 1/6 images from each) yields the best overall result. This suggests combining datasets may be a simple strategy to reduce dataset bias.

## 6 CONCLUSION

We revisit the dataset classification problem in the context of modern neural networks and large-scale datasets. We observe that the datasets bias can still be easily captured by modern neural networks. This phenomenon is robust across models, dataset combinations, and many other settings.

It is worth pointing out that the concrete forms of the bias captured by neural networks remain largely unclear. We have discovered that such bias may contain some generalizable and transferrable patterns, and that it may not be easily noticed by human beings. We hope further effort will be devoted to this problem, which would also help build datasets with less bias in the future.

---

[2]In this comparison, we search for the optimal learning rate, training epoch, and the layer from which the feature is extracted, following the common practice in the self-supervised learning community.

**Acknowledgements.** We thank Yida Yin, Mingjie Sun, Saining Xie, Xinlei Chen, and Mike Rabbat for valuable discussions and feedback, and all volunteers for participating in our user study.

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

## A    USER STUDY

To have a better sense of the dataset classification task, we further conduct a user study to assess how well humans can do this task and to learn their experience.

**Settings.** We ask our users to classify individual images sampled from the YCD combination. Because users may not be familiar with these datasets, we provide an interface for them to *unlimitedly* browse the training images (with ground-truth labels of their dataset identities) when they attempt to predict every validation image. We ask each user to classify 100 validation images, which do not overlap with the training set provided to them. We do not limit the time allowed to be spent on each image or on the entire test.

**Users.** A group of 20 volunteer participants participated in our user study. All of them are researchers with machine learning background, among which 14 have computer vision research experience.

**User study results.** Figure 5 shows the statistics of the user study results on the dataset classification task. In summary, 11 out of all 20 users have 40%-45% accuracy, 7 users have 45%-50%, and only 2 users achieve over 50%. The mean is 45.4% and the median is 44%.

The human performance is higher than the chance-level guess (33.3%), suggesting that there exist patterns that humans can discover to distinguish these datasets. However, the human performance is much lower than the neural network's 84.7%.

We also report that the 14 users who have computer vision research experience on average perform no better than the other users. Among these 14 users, we also ask the question "What accuracy do you expect a neural network can achieve for this task?" The estimations are 60% from 2 users, and 80% from 6 users, and 90% from 1 user; there were 5 users who chose not to answer. The users made these estimations before becoming aware of our work.

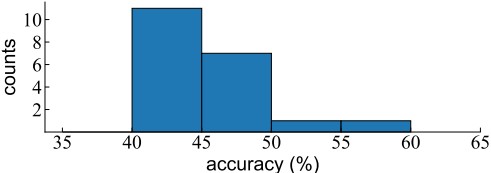

Figure 5: **User study results** on humans performing the dataset classification task. Humans generally categorize images from YCD with 40-60% accuracy.

There are 15 participants who describe the difficulty of the task as "difficult". No participant describes the task as "easy". 2 participants commented that they found the task "interesting".

We further asked the users what dataset-specific patterns they have used to solve this task. We summarize their responses below, in which brackets indicate how many users mentioned each pattern:

- YFCC: people (6), scenery (3), natural lighting, plants, lifestyle (2), real-world, sport, wedding, high resolution (2), darker, most specific, most new, cluttered;
- CC: cartoon (2), animated, clothing sample, product, logo, concept, explanatory texts, geography, furniture, animals, low resolution, colorful, brighter, daily images, local images, single person, realistic, clean background;
- DataComp: white background (3), white space, transparent background, cleaner background, single item (2), product (2), merchandise, logo-style, product showcase, text (2), lots of words, artistic words, ads, stickers, animated pictures (2), screenshots, close-up shot, single person, people, non-realistic icons, cartoon, retro.

In these user responses, there are some simple types of bias that can be exploited (*e.g.*, "white background" for DataComp), which can help increase the user prediction accuracy over chance-level guess. However, many types of the bias, such as the inclusion of "people" in images, are not meaningful for identifying the images (*e.g.*, all datasets contain images with people presented).

## B    IMPLEMENTATION DETAILS

For image-text datasets (CC, DataComp, WIT, LAION), we only use their images. The LAION dataset was filtered before usage. We uniformly sample the same number of images from each dataset

to form the training / validation sets for dataset classification. If a dataset already has pre-defined training / validation splits, we only sample from its train split. 1M images for each dataset is used as the default unless otherwise specified. This is not a small collection, yet it still only represents a tiny portion of images (e.g., <10%) for most datasets we study. To speed up image loading, the shorter side of each image is resized to 500 pixels if the original shorter side is larger than this. We observe this has minimal effect on the performance of models.

We train the models for the same number of samples seen as in a typical 300-epoch supervised training on ImageNet-1K classification (Liu et al., 2022), regardless of the number of training images. This corresponds to the same number of iterations as in Liu et al. (2022) since the same batch size is used. The complete training recipe is shown in Table 10.

| config | value |
|---|---|
| optimizer | AdamW |
| learning rate | 1e-3 |
| weight decay | 0.3 |
| optimizer momentum | $\beta_1, \beta_2{=}0.9, 0.95$ |
| batch size | 4096 |
| learning rate schedule | cosine decay |
| warmup epochs | 20 (ImageNet-1K) |
| training epochs | 300 (ImageNet-1K) |
| randomaug (Cubuk et al., 2020) | (9, 0.5) |
| label smoothing | 0.1 |
| mixup (Zhang et al., 2018b) | 0.8 |
| cutmix (Yun et al., 2019) | 1.0 |

Table 10: Training settings for dataset classification.

For the linear probing experiments on ViT-B in Section 5.3 and 5.4, we follow the settings used in MAE (He et al., 2022). For Section 5.4, we use the checkpoint from epoch 250, and sweep for a base learning rate from {0.1, 0.2, 0.3}, and a layer index for extracting features from {8, 9, 10}. For the cross-dataset generalization experiment in Section 5.5, we use a batch size of 1024 and keep all other hyper-parameters the same as those in MoCo v3 (Chen et al., 2021)

During inference, an image is first resized so that its shortest side is 256 pixels, maintaining the aspect ratio. Then the model takes its 224×224 center crop as input. Therefore, the model cannot directly exploit the different distributions of resolutions and/or aspect ratios for different datasets as a shortcut for predicting images' dataset identities. The model takes randomly augmented crops of 224×224 images as inputs in training.

## C ADDITIONAL RESULTS

**Training Curves.** We plot the training loss and validation accuracy for ConvNeXt-T YCD classification in Figure 6. The training converges quickly to a high accuracy level in the initial phases. This again demonstrates neural networks' strong capability in capturing dataset bias.

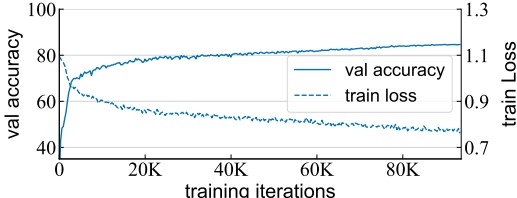

Figure 6: **Training curves for YCD classification.** The model converges quickly.

**ImageNet vs. ImageNetV2.** ImageNetV2 (Recht et al., 2019) attempts to create a new validation set trying to follow the exact collection process of ImageNet-1K's validation set. As such, the images look very much alike. We find a classifier could reach 81.8% accuracy classifying ImageNetV2 and ImageNet-1K's validation set, substantially higher than 50%, despite only using 8K images for training from each. This again demonstrates how powerful neural networks are at telling differences between seemingly close image distributions.

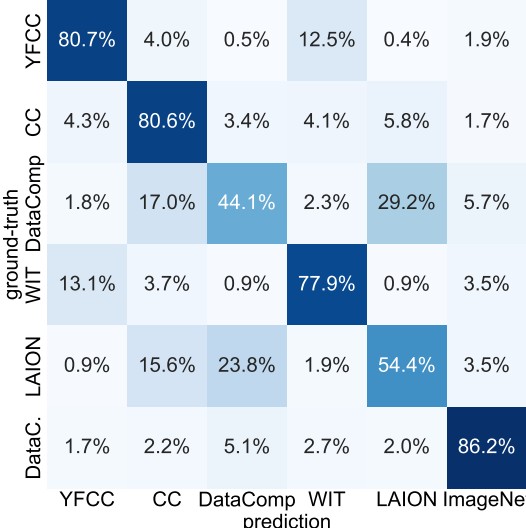

Figure 7: Confusion matrix for the 6-way classification in Table 2.

**Confusion Matrix.** We plot the confusion matrix for the 6-way dataset classification. We observe there exists a high confusion between DataComp and LAION. This is likely because they (Schuhmann et al., 2022; Gadre et al., 2023) both source from Common Crawl and apply filtering to select images that align closely with their captions in the CLIP (Radford et al., 2021) embedding space.

**Cross-dataset Generalization with MAE.** In addition to contrastive learning (MoCo v3) task used in Section 5.5, here we explore cross-dataset generalization problem with another surrogate task, masked autoencoding (MAE) (He et al., 2022). Table 11 shows the results. Similar to our previous observations, there is a clear diagonal pattern with the lowest validation loss in each column, although the differences in loss values in each column are much smaller than that in contrastive learning. This indicates that the model pre-trained on a given dataset only generalizes well within the same dataset but less well when transferred to others.

| train / eval | YFCC | CC | DataComp | WIT | LAION | ImageNet | average |
|---|---|---|---|---|---|---|---|
| YFCC | **0.419** | 0.394 | 0.320 | 0.434 | 0.332 | 0.397 | 0.383 |
| CC | 0.423 | **0.386** | 0.311 | 0.433 | 0.320 | 0.395 | 0.378 |
| DataComp | 0.428 | 0.393 | **0.306** | 0.437 | 0.317 | 0.394 | 0.379 |
| WIT | 0.423 | 0.394 | 0.317 | **0.427** | 0.328 | 0.396 | 0.381 |
| LAION | 0.429 | 0.392 | 0.306 | 0.439 | **0.314** | 0.395 | 0.379 |
| ImageNet | 0.425 | 0.395 | 0.312 | 0.437 | 0.325 | **0.389** | 0.380 |
| combined | 0.422 | 0.388 | 0.306 | 0.430 | 0.317 | 0.391 | **0.376** |

Table 11: **Cross-dataset generalization with MAE validation losses. Bold** indicates the lowest for each evaluation dataset (column).

# D    LIMITATIONS

While we have discovered that modern neural networks can achieve an excellent accuracy in classify which dataset an image is from, it is still unclear what are the exact forms of bias captured by neural networks. This warrants future research on understanding and interpreting the concrete forms of bias. In addition, this work examines a limited set of six large-scale image datasets, while leaving out many other popular image datasets, as well as datasets on other domains like videos and language.

