# OpenReview forum: "A Decade's Battle on Dataset Bias: Are We There Yet?"
_ICLR.cc/2025/Conference — ICLR 2025 Oral_

### Official Review · Reviewer_uFB6 · 2024-10-23

**Soundness:** 3
**Presentation:** 4
**Contribution:** 3
**Rating:** 8
**Confidence:** 4

**Summary:**

The paper revisits the "dataset classification" experiment from Torralba and Efros 2011 and applies it to modern neural networks and web-scale datasets. The authors find out that modern deep networks are very capable of solving such an artificial task, with a recognition accuracy of up to ~90%.

**Strengths:**

1) The paper's main finding, i.e., that the accuracy for the dataset classification task can be as high as 90% is surprising considering the size and diversity of the experimented web-scale datasets.

2) The authors perform a comprehensive set of experiments to cover multiple settings and ensure the validity of their findings, including 20 dataset combinations, several model sizes, architecture types, data augmentation strengths, and the number of available data points.

3) In Section 5, the paper provides additional analyses that rule out the possibility of achieving such high accuracy due to low-level signatures. Through a pseudo-dataset experiment and self-supervised learning, the authors also demonstrate that the networks are indeed learning dataset-related "semantic" features, further supporting their findings.

4) Overall, I found the paper well-written, clear, and easy to follow.

**Weaknesses:**

1) The paper is repeating an experiment that was introduced roughly 10 years ago but at a larger scale. Hence, despite the surprising practical result, the paper does not introduce a novel concept in absolute terms, and it does not propose a method to mitigate/decrease dataset bias (only observations and not practical solutions).

2) The paper lacks (even controlled) experiments to offer a qualitative evaluation of potential biases present in large-scale modern datasets. Looking at Table 2 (top panel), the classification performance significantly varies up to 30 percentage points.

3) Although the authors repeatedly assert, with empirical backing, that the networks can learn "semantic" features for dataset classification, they do not characterize these features in practice, even qualitatively. This absence of practical insight leaves a gap in understanding how the networks perform these tasks.

See Questions for possible improvements and suggestions.

**Questions:**

- Can a qualitative analysis be provided to explain why specific dataset classification tasks are significantly more challenging than others?
For instance, the ConvNeXt-T model on YFCC-CC-ImageNet scores 92.7%, whereas on CC-DataComp-LAION, it only reaches 62.8% (Table 2, top panel). Is this large discrepancy solely due to differences in distribution overlap? How could this phenomenon be visualized or understood more intuitively?

- It would also be interesting to visualize the difference in features learned by the network via, e.g., intra-class visualizations in latent space [1] or CAM approaches, compared to features learned when performing an actual classification task.

[1] Donglai Wei, Bolei Zhou, Antonio Torrabla, and William Freeman; Understanding Intra-Class Knowledge Inside CNN

Overall, the paper's strengths, including the finding of high dataset classification accuracy at large scales and the extensive, well-executed experiments, outweigh its weaknesses.
The findings interest the community, and the work is clear and well-presented. While addressing the identified weaknesses, such as providing more qualitative insights into the learned features and biases, which would strengthen the contribution, I am leaning towards acceptance. I would be inclined to raise the score if these aspects are improved.

---POST REBUTTAL---

The authors provided an extensive rebuttal covering my main concerns. For this reason, I am updating the score. I encourage to include the additional analyses performed in the final manuscript to strengthen the contribution.

---

> ### Author Response · Authors · 2024-11-25
> **Rebuttal by Authors**
>
> We are glad that you find our experiments to show the dataset bias are very comprehensive. We address your concerns and questions below:
>
> > W1 (1): The paper is repeating an experiment that was introduced roughly 10 years ago but at a larger scale. Hence, despite the surprising practical result, the paper does not introduce a novel concept in absolute terms.
>
>
> **We agree with you that our work is a revisit of Torralba & Efros [1].  However, the context has changed a lot over a decade, and we do believe this is a timely visit to an important issue whose degree may not be fully appreciated in the community.**
>  - The datasets examined in [1] are understandably biased in retrospect, as they are mostly specialized and the differences are very easily recognizable (e.g., Caltech101, SUN), yet the currently popular datasets we examine are large-scale and supposedly much more diversified. Before we do the experiments, it is hard to tell the exact classification results accurately.
> - In addition, deep learning and neural networks were not considered in [1], which was before the deep learning revolution started in 2012. Neural Networks may entirely change how much bias can be captured compared to traditional classifiers.
>
> **Our primary objective is to raise awareness of building more diverse and unbiased pretraining vision datasets, and prompt further research in developing approaches for this, such as quantitative tools for measuring dataset bias.** Without a collective recognition of the degree of bias present in current large-scale datasets, there may not be enough motivation to mitigate them.
>
> [1] Torralba & Efros. Unbiased Look at Dataset Bias. CVPR’11
>
> &nbsp;
>
> > W1 (2): It does not propose a method to mitigate/decrease dataset bias (only observations and not practical solutions)
>
> **For methods to mitigate bias, we find simply combining multiple datasets and using the combined one to train the model to be a straightforward mitigation.** We have illustrated the potential of this in Table 9 of Section 5.5. Specifically, we conduct cross-dataset training and validation using masked autoencoding (MAE [1]) training. Each value entry is the validation loss for each validation dataset (column). We can see in the last row, that simply combining all datasets yields the best overall result, while also achieving the second best for each individual evaluation dataset.
>
> [1] He et al. Masked Autoencoders Are Scalable Vision Learners. CVPR’22
>
> &nbsp;
>
> > W2 (1): The paper lacks (even controlled) experiments to offer a qualitative evaluation of potential biases present in large-scale modern datasets.
>
> **We would like to note that Table 2 could be considered a controlled experiment when comparing any two rows that only differ by one dataset.** For example, the first and second rows (YFCC, CC, DataComp vs YFCC, CC, WIT) both contain YFCC and CC but are different in whether they use DataComp or WIT. Therefore, the difference in classification accuracy for these two rows can mainly be attributed to this dataset difference. **Furthermore, the bottom panel adds one dataset from the six datasets step by step. This could qualitatively illustrate how each dataset’s addition changes the classification accuracy.**
>
> **For more fine-grained control/isolation experiments, a direct follow-up of our work ([https://openreview.net/forum?id=NGIIHlAEBt](https://openreview.net/forum?id=NGIIHlAEBt)) based on our paper has explored the specific forms of bias present in large-scale datasets (e.g., YFCC, CC, and DataComp).** They use various transformations to isolate different types of information in each dataset and then perform the dataset classification on the transformed datasets. For example, they take a pre-trained semantic segmentation model to extract semantic segmentation maps for images in each dataset and then train a new classifier on these maps. **They found that transforming original images into semantic (e.g., semantic segmentation maps, object bounding boxes, and captions) and structure representations (e.g., object contours and depth maps) can still achieve a high dataset classification accuracy.**

---

> ### Author Response · Authors · 2024-11-25
> **Rebuttal by Authors**
>
> > W2 (2): Looking at Table 2 (top panel), the classification performance significantly varies up to 30 percentage points.
>
> **Regarding your question about the significant variance in classification results across different 3-way datasets, this likely stems from the high similarity between DataComp and LAION.** When we exclude rows in Table 2 involving both DataComp and LAION, all classification accuracies are higher than 80%. We also include the confusion matrix of 6-way classification in Table 2 here ([https://drive.google.com/file/d/1JU5C4hqSwsQ019moNxxOSdm2S7zR81j1](https://drive.google.com/file/d/1JU5C4hqSwsQ019moNxxOSdm2S7zR81j1)). We can see that DataComp and LAION images are confused far more often than other pairs.
>
> **To understand this confusion between DataComp and LAION more intuitively, we provide a 10x10 grid cell panel to visualize 100 random samples from each dataset here ([https://drive.google.com/file/d/1uCekv_Lwlj_-YFnbnpWRgWJrRNjM18-d](https://drive.google.com/file/d/1uCekv_Lwlj_-YFnbnpWRgWJrRNjM18-d)).** We can see both datasets contain many object-centric images of T-shirts and the front cover of the book. Their high similarity is likely due to two factors: (1) DataComp and LAION source their images from Common Crawl, and (2) both datasets apply filtering to select images that align closely with their captions in the CLIP embedding space.
>
>
>
> &nbsp;
>
>
> > W3: Although the authors repeatedly assert, with empirical backing, that the networks can learn "semantic" features for dataset classification, they do not characterize these features in practice, even qualitatively. This absence of practical insight leaves a gap in understanding how the networks perform these tasks.
>
> **We are glad that our work served as a starting point for the follow-up work ([https://openreview.net/forum?id=NGIIHlAEBt](https://openreview.net/forum?id=NGIIHlAEBt)), which has pinpointed various forms of these semantic features.** They leveraged Large Language Models (LLMs) to summarize dataset-specific patterns based on the generated captions from images. They found that YFCC contains abundant outdoor, natural, and human-related scenes, while DataComp concentrates on static objects and digital graphics with clean backgrounds and minimal human presence; in contrast, CC blends elements of both YFCC's dynamic scenes and DataComp's static imagery.
>
> Our work indeed did not provide a direct analysis of the semantic features, we are glad that it serves as a first step for raising awareness of dataset bias, and future work like ([https://openreview.net/forum?id=NGIIHlAEBt](https://openreview.net/forum?id=NGIIHlAEBt)) can build upon it to better understand the forms of bias.
>
>
> &nbsp;
>
> > Q1 (1): Can a qualitative analysis be provided to explain why specific dataset classification tasks are significantly more challenging than others? For instance, the ConvNeXt-T model on YFCC-CC-ImageNet scores 92.7%
>
> To understand the difference between YFCC, CC, and ImagenNet more intuitively, we follow language analysis from ([https://openreview.net/forum?id=NGIIHlAEBt](https://openreview.net/forum?id=NGIIHlAEBt)) and leverage Large Language Models (LLMs) to summarize each dataset-specific pattern in natural language descriptions. The results are shown here ([https://drive.google.com/file/d/1QBBq-gskodhMr-vTC_bfgm9cSS9pqK_g](https://drive.google.com/file/d/1QBBq-gskodhMr-vTC_bfgm9cSS9pqK_g)). **Each dataset has its own unique pattern. YFCC focuses on social gatherings and natural landscapes; CC features human interactions and domestic environments; and ImageNet is related to vibrant sporting activities and animal encounters.**
>
> &nbsp;
>
> > Q1 (2): CC-DataComp-LAION, it only reaches 62.8% (Table 2, top panel). Is this large discrepancy solely due to differences in distribution overlap? How could this phenomenon be visualized or understood more intuitively?
>
> We have addressed this question in W2 (2).
>
> &nbsp;
>
> > Q2: It would also be interesting to visualize the difference in features learned by the network via, e.g., intra-class visualizations in latent space or CAM approaches, compared to features learned when performing an actual classification task.
>
> Thank you for suggesting these visualization methods. Below, we apply the GradCAM [1] on the dataset classification model (trained on YFCC, CC, and DataComp) and an actual classification model (trained on ImageNet-1K) to highlight key regions in each image. The results are here ([https://drive.google.com/file/d/1vPpjHLNqceB7sDe3CA3cjdaeoSKeeq9M](https://drive.google.com/file/d/1vPpjHLNqceB7sDe3CA3cjdaeoSKeeq9M)). **We can see both models focus on semantic meaningful things.** They even sometimes attend to the same objects in each image, such as birds in the third image of YFCC and shoes in the first image of DataComp. Interestingly, the dataset classification model often attends to multiple regions within an image, while the ImageNet-1K model typically focuses on a single region.

---

> ### Author Response · Authors · 2024-11-25
> **Rebuttal by Authors**
>
> (Continued from Q2)
>
> [1] Selvaraju et al. Grad-cam: Visual explanations from deep networks via gradient-based localization. ICCV’17.
>
> &nbsp;
>
> We thank you for your valuable feedback and see resolving your questions and concerns as a great improvement to our paper. If you have any further questions or concerns, we are very happy to answer.

---

> > ### Comment · Reviewer_uFB6 · 2024-11-26
> >
> > I thank the authors for the extensive and detailed rebuttal, which includes several additional analyses that cover my initial concerns.
> >
> > For this reason, I will update my score

---

> > > ### Author Response · Authors · 2024-11-28
> > >
> > > Thank you for acknowledging the response and updating the review!

---

### Official Review · Reviewer_mF8E · 2024-11-03

**Soundness:** 4
**Presentation:** 3
**Contribution:** 3
**Rating:** 8
**Confidence:** 4

**Summary:**

The paper studies the phenomenon of dataset bias through the lens of dataset classification. In this work, the authors expand upon the "Name the dataset" experiment from Torralba and Efros (2011) to modern neural networks. They then observe that modern NNs are relatively successful at the task of dataset classification, that is, predicting the source dataset of an image, under a variety of settings. The claim is that dataset bias still exists as NNs can exploit minor differences between large-scale image datasets to successfully solve the problem. The authors further test upon their hypothesis by designing and evaluating ConvNext, ViTs, and other models under a variety of confounding factors.

**Strengths:**

1. The paper is very well written, and clearly conveys the experiments as well as the intuition behind each one.

2. The experiments are well-designed and clearly support the conclusions. While the underlying idea of `dataset classification' is not new, the authors have extended the experiments to clearly account for a variety of confounding factors.

3. It is also interesting to observe that the dataset bias task is also learning semantically useful features similar to pretraining, and self-supervised learning objectives. This is of independent interest as it can be used as a quick, cheaper method to initialize networks. While performance lags behind other approaches, this opens up some interesting ideas towards using dataset bias as an exploiting factor for self-supervision in future work.

4. The model generalization vs memorization experiments show a significant vulnerability in deep networks exploiting dataset-specific features, with methods like augmentation failing to correct for the same.

**Weaknesses:**

Overall, I found the paper to be well-written and clear. However, I am listing some points for improvement,

1.  It might be a useful exercise to use unbalanced mixes of the datasets as well, to observe effects of bias when a model encounters sources with varying numbers of images.

2. I also suggest adding confusion matrices for the experiments in Table 2 to help understand correlations between datasets.

3. The paper also does not address any directions on how to specifically tackle dataset bias. Given that standard approaches like data augmentation do not work, I am curious to know the authors' thoughts on methods like synthetic data generation, or domain generalization approaches as possible solutions to the problem.

**Questions:**

Please see weaknesses above.

---

> ### Author Response · Authors · 2024-11-25
> **Rebuttal by Authors**
>
> We are glad that you find our observation that dataset classification can learn semantic features interesting. We address your concerns and questions below:
>
> > W1: It might be a useful exercise to use unbalanced mixes of the datasets as well, to observe effects of bias when a model encounters sources with varying numbers of images.
>
> **We observe that when the model trained on unbalanced mixes of datasets has a lower dataset classification accuracy than the one trained on a balanced dataset with balanced sampling from each source.** Specifically, we created a training dataset consisting of 3M samples, with 50% sourced from YFCC, 25% from CC, and 25% from Datacomp. For validation, we maintained a regular 10K sample for each dataset. The model trained on this unbalanced mix of samples achieves an accuracy of 80.1%, which is 1.3% lower than the one trained on the balanced samples (81.4%).
> Below, we also provide confusion matrices comparing the results of balanced sampling and unbalanced sampling approaches here ([https://drive.google.com/file/d/1FSdl5A-fHAncrPSAH_0VoXDtI8AXsquO](https://drive.google.com/file/d/1FSdl5A-fHAncrPSAH_0VoXDtI8AXsquO)). The number on the first column of the confusion matrix with unbalanced sampling (right) is higher than the one with balanced sampling (left). This suggests that the model trained on unbalanced mixes of datasets tends to predict more images as YFCC regardless of the actual dataset identity.
>
> &nbsp;
>
> > W2: I also suggest adding confusion matrices for the experiments in Table 2 to help understand correlations between datasets.
>
> Thank you for your suggestion. **Due to the limited space in the main text, we have added the confusion matrix of the 6-way dataset classification (YFCC, CC, DataComp, WIT, LAION, and ImageNet) in Appendix C of our draft.** We also present here ([https://drive.google.com/file/d/1JU5C4hqSwsQ019moNxxOSdm2S7zR81j1](https://drive.google.com/file/d/1JU5C4hqSwsQ019moNxxOSdm2S7zR81j1)) for your convenience. Notably, DataComp and LAION images are confused far more often than other pairs. This is likely because they both sample from Common Crawl and apply filtering to select images that align closely with their captions in the CLIP embedding space.
>
> &nbsp;
>
> > W3: The paper also does not address any directions on how to specifically tackle dataset bias. Given that standard approaches like data augmentation do not work, I am curious to know the authors' thoughts on methods like synthetic data generation, or domain generalization approaches as possible solutions to the problem.
>
> **It is worth noting that dataset bias may be inherited from the generative models' training images to its generated images.** A follow-up study based on our work ([https://openreview.net/forum?id=NGIIHlAEBt](https://openreview.net/forum?id=NGIIHlAEBt)) demonstrated this point. Specifically, they train a diffusion model on each of the datasets YFCC, CC, and DataComp (YCD) and then generate new images from each of the models. They find the dataset classification accuracy on these synthetic images is very close to the one on the original images from YCD dataset. We think how to use synthetic images but with less bias from their training data is an important problem to study.
>
> **More broadly, synthetic data can be very useful to augment existing datasets.** Numerous studies have demonstrated its effectiveness in reducing dataset bias across various domains, including facial recognition [1, 2, 3], medical imaging [4], and other visual datasets [5, 6, 7]. It would be very valuable to further extend their approaches to augment large-scale visual datasets in pre-training and see whether they are helpful in reducing dataset bias.
>
> **Leveraging domain generalization techniques to reduce dataset bias is a promising research direction.** Various studies [8, 9, 10] have explored methods to mitigate domain bias in fully supervised settings with annotated labels. More recently, the domain generalization technique [11] has also shown effectiveness on unlabelled data. We are very interested in whether these domain generalization approaches can reduce bias for datasets scaled up to millions (or billions). We aim to explore this direction in future research.
>
> **In our view, combining multiple datasets might be a straightforward approach to reducing dataset bias.** We actually have shown the potential of this in Table 9 of Section 5.5, where we conduct cross-dataset training and validation using masked autoencoding (MAE [12]) training. When trained with multiple datasets, the model can achieve the lowest average transfer loss.  It also achieves the second best for each individual evaluation dataset. This suggests combining datasets is a simple strategy to reduce bias.

---

> ### Author Response · Authors · 2024-11-25
> **Rebuttal by Authors**
>
> (Continued from W3)
>
> [1] Ramaswamy et al. Fair attribute classification through latent space de-biasing. CVPR’21\
> [2] Kortylewski et al. Analyzing and reducing the damage of dataset bias to face recognition with synthetic data." ICCV workshop’19\
> [3] Sharmanska et al. Contrastive examples for addressing the tyranny of the majority. arXiv’20\
> [4] MaayanFrid et al. Gan- based synthetic medical image augmentation for increased cnn performance in liver lesion classification. Neurocomputing’18\
> [5] Li et al. Bigdatasetgan: Synthesizing imagenet with pixel-wise annotations. CVPR’22\
> [6] Dunlap et al. Diversify your vision datasets with automatic diffusion-based augmentation. NeurIPS’23\
> [7] Azizi et al. Synthetic data from diffusion models improves imagenet classification. arXiv’23\
> [8]  Motiian et al. Unified deep supervised domain adaptation and generalization. ICCV’17\
> [9] Seo et al. Learning to optimize domain specific normalization for domain generalization. ECCV’20\
> [10] Zhao et al. Domain generalization via entropy regularization. NeurIPS’20\
> [11] Zhang et al. Towards unsupervised domain generalization. CVPR’22\
> [12] He et al. Masked Autoencoders Are Scalable Vision Learners. CVPR’22
>
>
>
> &nbsp;
>
> We thank you for your valuable feedback and addressing your questions and concerns is a great improvement to our paper. If you have any further questions or concerns, we are very happy to answer.

---

> > ### Comment · Reviewer_mF8E · 2024-11-27
> > **Thank you**
> >
> > I thank the authors for their responses, and as before, find the paper to be a valuable and interesting contribution to the field. It is also interesting to see that mixing datasets tends to offset some of the bias. I also appreciate the authors' thoughts on the use of synthetic data, and domain generalization. Overall, this is a good paper, and I am keeping my positive score.

---

> > > ### Author Response · Authors · 2024-11-28
> > >
> > > Thank you for the feedback and the appreciation!

---

### Official Review · Reviewer_Bad9 · 2024-11-04

**Soundness:** 4
**Presentation:** 3
**Contribution:** 3
**Rating:** 8
**Confidence:** 4

**Summary:**

This paper presents a modern version of the dataset classification experiment proposed by [Torralba & Efros (2011)](https://people.csail.mit.edu/torralba/publications/datasets_cvpr11.pdf): training a neural network to assign images to the right dataset it comes from. SVM reached reasonable accuracies by then on image classification datasets -- indicating dataset bias exist.

This paper experiments with modern backbones (eg, ConvNeXt and ViT) on modern large-scale datasets (eg, WIT, LAION), finding that dataset classification becomes even easier -- dataset bias is easier to spot by models despite the datasets being diverse and large-scale. The authors tried to explain this phenomenon, but it turns out the bias is not attributed to low-level clues and models do not simply memorize the data -- they learn high-level signals. The authors also show that self-supervised representations -- which tend to be good in semantics -- also work well on dataset classification, indicating that the dataset bias can be attributed to some high-level pattern.

**Strengths:**

- This paper presents a timely revisiting of an old question: how biased are ML datasets in the view of ML models? While it is widely accepted that some modern datasets are substantially diverse and large-scale, they can still be easily discriminated by modern models. This is a fresh conclusion and could trigger meaningful rethinking about the bias of large-scale datasets and models trained upon.

- The study is comprehensive and well-presented. It covers most modern backbones and pertaining datasets, and studies various factors that affect dataset classification performance. It is also accompanied by a thorough analysis that considers what kind of clue networks are used to discriminate these datasets, what kind of representation is beneficial, and how well models generalize across datasets, etc.

**Weaknesses:**

The way this paper is presented might be over-focused on the dataset classification task itself, and it might require readers to read [Torralba & Efros (2011)](https://people.csail.mit.edu/torralba/publications/datasets_cvpr11.pdf) first to gain a full picture. In the context of that paper, the task was proposed to show that image classification datasets by then are biased and cannot make a complete representation of the visual world. They discussed possible sources of such bias (eg, in data selection), but more importantly it also led to discussions on how to measure the value of a dataset (cross-dataset generalization), how such biases shape representations (eg, different negative sets make models capture different aspects of representations), and how to overcome such biases (eg, random cropping augmentation). If without considerable discussions on the latter, it might be unclear for readers to understand the significance of studying the dataset classification task.

**Questions:**

- To me there might never be a simple answer to the question of what kind of signal models learn when they succeed in discriminating the dataset one image comes from. Modern neural networks are so good at overfitting and any subtle patterns in dataset curation -- no matter low-level or high level, could be captured by them and processed to a strong black-box representation. If we accept that there is bias between CC, WIT, and LAION, and the bias remains as these datasets grow in scale, then what kind of harm would there be? If the representations learned from them generalize well, is the bias no longer a concern?

- When we only consider datasets at a very large scale (eg, >100M), will it still be easy for models to tell the difference?

- Dataset classification may not be a universal indicator of the bias between datasets, as it only sees one sample each time. Other aspects like concept frequency distribution (motivation of MetaCLIP) are also an important aspect of dataset bias and is highly related to the quality of representations learned upon. Do we need some new surrogate task/indicator for measuring bias in modern datasets in the next decade?

---

> ### Author Response · Authors · 2024-11-25
> **Rebuttal by Authors**
>
> We are glad that you find our paper revisits the timely and critical question of dataset bias in machine learning and offers fresh insights into how even modern, diverse, and large-scale datasets are easily distinguishable by state-of-the-art models. We address your concerns and questions below:
> > W1: The way this paper is presented might be over-focused on the dataset classification task itself, and it might require readers to read Torralba & Efros (2011) first to gain a full picture. In the context of that paper, the task was proposed to show that image classification datasets by then are biased and cannot make a complete representation of the visual world. They discussed possible sources of such bias (eg, in data selection), but more importantly it also led to discussions on how to measure the value of a dataset (cross-dataset generalization), how such biases shape representations (eg, different negative sets make models capture different aspects of representations), and how to overcome such biases (eg, random cropping augmentation). If without considerable discussions on the latter, it might be unclear for readers to understand the significance of studying the dataset classification task.
>
> **We have added one paragraph at the beginning of the introduction to aid the readers who do not have too much context on Torralba & Efros’s work [1] and to stress its significance.** We paste it here for your convenience:
>
> "In 2011, Torralba & Efros (2011) called for a battle against dataset bias in the community, right before the dawn of the deep learning revolution [2]. They introduced the *Name That Dataset* experiment, where images are sampled from each of several datasets, and a model is trained on the union of these images to classify which dataset an image is taken. Remarkably, datasets at that time could be classified with high accuracy. They also found that a model trained on one dataset can only perform well on that dataset but fails to generalize to others."
>
> [1] Torralba & Efros. Unbiased Look at Dataset Bias. CVPR’11\
> [2] Krizhevsky et al. Imagenet Classification with Deep Convolutional Neural Networks. NeurIPS’12
>
> &nbsp;
>
> > Q1 (1): To me there might never be a simple answer to the question of what kind of signal models learn when they succeed in discriminating the dataset one image comes from. Modern neural networks are so good at overfitting and any subtle patterns in dataset curation -- no matter low-level or high level, could be captured by them and processed to a strong black-box representation. If we accept that there is bias between CC, WIT, and LAION, and the bias remains as these datasets grow in scale, then what kind of harm would there be?
>
> **Models pre-trained on one biased dataset are likely to reflect the dataset's bias, making them less robust and unreliable in various real-world scenarios.** A direct follow-up study of our paper ([https://openreview.net/forum?id=NGIIHlAEBt](https://openreview.net/forum?id=NGIIHlAEBt)) conducted analysis from many perspectives to isolate the concrete forms of bias, and on a very high level, summarized that “YFCC emphasizes outdoor and natural scenes with human interactions, while DataComp features digital graphics heavily.” Therefore, it is plausible for the model pre-trained on YFCC to perform well on outdoor imagery, but struggle with tasks involving digital graphics.
>
> &nbsp;
>
> > Q1 (2): If the representations learned from them generalize well, is the bias no longer a concern?
>
> **In our paper, we do find the bias in a dataset can affect the learned presentations and consequently compromise the model’s ability to generalize across different datasets.** In Table 9 of Section 5.5, we conduct cross-dataset training and validation using masked autoencoding (MAE [1]) training. We find the model only performs the best when tested on the pretraining dataset, but generalizes less well to other datasets. This suggests that the representation learned on one dataset might not transfer at 100% effectiveness to other datasets.
>
> **However, we agree with you that it is indeed hard to tell whether such a degree of performance compromise matters.** Learning transferable representations and being able to generalize is indeed the wonder of deep learning. We think testing with more out-of-distribution and more specialized downstream datasets like medical images may be able to demonstrate the effect of bias better.
>
>
> [1] He et al. Masked Autoencoders Are Scalable Vision Learners. CVPR’22

---

> ### Author Response · Authors · 2024-11-25
> **Rebuttal by Authors**
>
> > Q2: When we only consider datasets at a very large scale (eg, >100M), will it still be easy for models to tell the difference?
>
> We'd like to note that we only take 1M images from each dataset as the training set  in most of our experiments. **As a result, what matters more for the classification accuracy is the distribution the dataset represents, not directly its scale.** If we could expand datasets to be 1000x of their sizes, but still at the same image distributions, the classification accuracy would not change much.
>
> Nonetheless, YFCC, DataComp, and LAION are three datasets we experimented that have 100M or more images (100M, 1B, 2B images respectively). In Table 2 of our paper, the model trained on these datasets achieves a dataset classification accuracy of only 62.8%, indeed the lowest among all possible 3-way classifications. **This relatively low accuracy can likely be attributed to two factors: (1) DataComp and LAION both source their images from Common Crawl, and (2) DataComp and LAION's curation methods are similar in that they both apply filtering to select images that align closely with their captions in the CLIP embedding space.**
>
> We want to emphasize that scale alone should not change the classification accuracy, and it is the dataset distributions that matter. **However, scales could confound with distributions - very large scale (billion) datasets' distributions may unavoidably become similar to each other, as only a (roughly) "Internet distribution" source like Common Crawl can provide so many images.**
>
> &nbsp;
>
> > Q3: Dataset classification may not be a universal indicator of the bias between datasets, as it only sees one sample each time. Other aspects like concept frequency distribution (motivation of MetaCLIP) are also an important aspect of dataset bias and are highly related to the quality of representations learned upon. Do we need some new surrogate task/indicator for measuring bias in modern datasets in the next decade?
>
> **We agree with you that dataset classification might not be a good universal indicator for dataset bias. A key limitation is that this approach needs multiple datasets to begin with, and cannot give a quantitative evaluation given only a single pretraining dataset.** To better measure dataset bias in general, we could consider the following approach:
> 1. Pretrain a self-supervised method (e.g., MAE [1] or MoCo v3 [2]) on each pretraining dataset.
> 2. Then, evaluate the performance on a range of downstream datasets by either:
>     - Measuring the transfer loss on each downstream dataset, or
>     - Linear probing the features learned from the pre-trained model on each downstream dataset.
>
> **The average transfer loss or linear probing accuracy across these downstream tasks can then serve as a proxy metric for the bias of the pretraining dataset.** A lower transfer loss (or higher linear probing accuracy) would indicate the learned representations from the pre-training dataset can help the model generalize across diverse tasks and thus the dataset itself is less biased. The key is to select a diverse and representative set of downstream datasets. We aim to explore this direction in future research.
>
> [1] He et al. Masked Autoencoders Are Scalable Vision Learners. CVPR’22\
> [2] Chen et al. An Empirical Study of Training Self-Supervised Vision Transformers. ICCV’21
>
> &nbsp;
>
> We thank you for your valuable feedback and addressing your questions and concerns is a great improvement to our paper. If you have any further questions or concerns, we are very happy to answer.

---

> > ### Comment · Reviewer_Bad9 · 2024-11-27
> >
> > Thanks to the authors for their detailed response and I confirm my positive recommendation.

---

> > > ### Author Response · Authors · 2024-11-28
> > >
> > > Thanks for the feedback and the acknowledgement!

---

### Meta-Review · Area_Chair_PwvH · 2024-12-08

**Metareview:**

The paper revisits the classic dataset classification experiment by Torralba & Efros (2011) using modern, large-scale datasets and state-of-the-art neural network architectures. It demonstrates that despite the diversity and scale of current datasets, dataset bias persists, as evidenced by high classification accuracy across a range of configurations. The study's findings challenge the assumption that modern datasets are unbiased and highlight the need for the community to address these biases systematically.

Strengths:
* Timely Contribution: The paper revisits a critical topic, providing fresh insights into the extent of bias in large-scale, diverse datasets.
* Comprehensive Experiments: Includes evaluations across various datasets, model architectures, and configurations, ensuring robustness of the findings.
* Novel Observations: Demonstrates that dataset classification tasks reveal high-level semantic biases, which align with self-supervised learning representations.
* Practical Implications: Suggests that combining datasets and addressing bias can improve model generalization across tasks.

Weaknesses:
* Lack of Mitigation Methods: While the paper identifies biases, it does not propose novel approaches to mitigate them, though follow-up studies and potential directions are discussed.
* Limited Qualitative Analysis: More in-depth characterization of the learned semantic features and qualitative biases could strengthen the understanding of dataset distinctions.

Overall, this paper offers a significant and timely contribution to understanding dataset bias in the deep learning era. It effectively raises awareness and lays a solid foundation for future research on mitigating biases in large-scale datasets. The clarity, soundness, and potential impact of the findings justify its acceptance.

**Additional Comments On Reviewer Discussion:**

Summary of Rebuttal Discussion and Changes:

Focus on Dataset Classification Task (Raised by Bad9):
* Concern: Paper focuses heavily on dataset classification without broader context.
* Response: Authors added a detailed introduction linking to Torralba & Efros (2011) and expanded on the broader implications of dataset bias.


Qualitative and Confusion Analyses (Raised by uFB6 and mF8E):
* Concern: Lack of qualitative evaluation and confusion matrices.
* Response: Authors provided confusion matrices, visualizations, and qualitative analyses of dataset distinctions using GradCAM and semantic features.

Lack of Mitigation Strategies (Raised by uFB6 and mF8E):
* Concern: No solutions to reduce dataset bias proposed.
* Response: Authors proposed combining datasets, leveraging synthetic data, and domain generalization as future directions, supported by cross-dataset validation.

Repetition of Prior Experiment (Raised by uFB6):
* Concern: Experiment revisits an old problem without introducing new concepts.
* Response: Authors emphasized the importance of revisiting this topic in the modern context with new datasets and architectures.

---

### Decision · Program_Chairs · 2025-01-22

Accept (Oral)